# Can Baseline IL-6 Levels Predict Long COVID in Subjects Hospitalized for SARS-CoV-2 Disease?

**DOI:** 10.3390/ijms24021731

**Published:** 2023-01-15

**Authors:** Lydia Giannitrapani, Luigi Mirarchi, Simona Amodeo, Anna Licata, Maurizio Soresi, Francesco Cavaleri, Salvatore Casalicchio, Gregorio Ciulla, Maria Elena Ciuppa, Melchiorre Cervello, Mario Barbagallo, Nicola Veronese

**Affiliations:** 1Department of Health Promotion, Mother and Child Care, Internal Medicine and Medical Specialties, University of Palermo, 90133 Palermo, Italy; 2Institute for Biomedical Research and Innovation, National Research Council, 90146 Palermo, Italy

**Keywords:** inflammation, COVID-19, long COVID, IL-6

## Abstract

The immune response to infection plays a crucial role in the pathogenesis of COVID-19, but several patients develop a wide range of persistent symptoms, which is becoming a major global health and economic burden. However, reliable indicators are not yet available to predict the persistence of symptoms typical of the so-called long COVID. Our study aims to explore an eventual role of IL-6 levels as a marker of long COVID. Altogether, 184 patients admitted to the COVID Medicine Unit of the University Hospital in Palermo, Italy, from the 1st of September 2020, were analyzed. Patients were divided into two groups according to the IL-6 serum levels (normal or elevated), considering the serum IL-6 levels measured during the first four days of hospitalization. In our study, higher serum IL-6 levels were associated with a doubled higher risk of long COVID (OR = 2.05; 95% CI: 1.04–4.50) and, in particular, they were associated with a higher incidence of mobility decline (OR = 2.55; 95% CI: 1.08–9.40) and PTSD (OR = 2.38; 95% CI: 1.06–8.61). The analysis of our case series confirmed the prominent role of IL-6 levels in response to SARS-CoV-2 infection, as predictors not only of COVID-19 disease severity and unfavorable outcomes, but also long COVID development trends.

## 1. Introduction

Coronavirus-19 disease (COVID-19) has several clinical manifestations, ranging from completely asymptomatic to mild, moderate, severe, and rapidly progressive forms that can lead to multiorgan failure and death; but, to date, although the physio-pathogenetic mechanisms underlying this evolution are fairly well known, there are no validated markers capable of predicting the behavior of the disease.

In particular, a subset of patients after an acute SARS-CoV-2 infection develops a wide range of persistent symptoms [1,2,3] and a recent study on COVID-19 patients who were followed for 9 months after the acute phase found that approximately 30% reported such persistent symptoms [4]. For these patients, a diagnosis of long COVID, post-acute COVID-19 syndrome, or post-acute sequelae of COVID-19 (PACS) has been coined.

Constant fatigue, cognitive impairment (“brain fog”/forgetfulness/poor concentration), headache, dyspnoea, arthralgia/myalgia, sleep disturbance, depression/anxiety, palpitations, chronic rhinitis, dysgeusia, and sore throat are among the main symptoms that, if lasting more than 60 days after the acute infection, shape the diagnosis of long COVID [1,2,3].

The inflammatory response is known to play a critical role in COVID-19 disease progression, and the so-called “cytokine storm” is crucial and can lead to the development of the most serious complications and death [5].

It is well known that many cytokines take part in the cytokine storm in COVID-19 patients, including IL-6, IL-1, IL-2, IL-10, TNF-α, and IFN-γ [6]; however, a crucial role seems to be played by IL-6, whose increased serum levels have been correlated with respiratory failure, acute respiratory distress syndrome (ARDS), and adverse clinical outcomes [7]. Some studies indicated that IL-6 levels may be used to predict the occurrence of severe COVID-19 [8], respiratory failure [9], or long-term neuropsychiatric symptoms of COVID-19 [10].

During the evolution of the COVID-19 clinical picture, the secretion of multiple cytokines is closely related to the development of clinical symptoms and, in particular, IL-6 can be involved into vascular leakage, the activation of complement, and the coagulation cascade, leading to the characteristic symptoms of severe forms, such as those that can develop a diffuse intravascular coagulation (DIC) [11,12]. IL-6 is also likely to cause cardiomyopathy by promoting myocardial dysfunction, which is often observed in patients with critical forms of COVID-19 [13]. In addition, the activation of endothelial cells and endothelial dysfunction can lead to capillary leakage, hypotension, and coagulopathy [14].

IL-6 is an interleukin that acts as a multifunctional cytokine. It is secreted by T lymphocytes and macrophages to stimulate the immune response during an infection. It has significant pro-inflammatory properties and pleiotropic effects on the acquired immune system (B and T cells) and on the innate immune system (neutrophils, macrophages, and natural killer cells) [15]. The actions of IL-6 are mediated through binding to soluble or membrane-bound IL-6 receptors (IL-6R, gp80) [16], which in turn induce the interaction of another cell surface polypeptide chain called gp130, expressed in almost all tissues and cells, some of which do not express the 80-kDa IL-6R, thus explaining the pleiotropy and redundancy of the IL-6 family of cytokines [17].

It has been hypothesized that long COVID could be the consequence of persistent reservoirs of SARS-CoV-2 in certain tissues, the re-activation of other neurotrophic pathogens under conditions of COVID-19 immune dysregulation, SARS-CoV-2 interactions with host microbiome/virome communities, clotting/coagulation issues, disrupted brainstem/vagus nerve signaling, ongoing activity of primed immune cells, and autoimmunity due to molecular mimicry between pathogen and host peptides [18].

Considering that the role of IL-6 in the acute phase of the disease is well established [19], the aim of our study was to evaluate if there could also be a role of IL-6 in predicting the development of a long COVID condition, after hospital discharge.

## 2. Results

The 184 patients had a mean age of 62.1 (range: 17–89) years and they were prevalently males (52.5%). The median serum IL-6 level was 13.4 pg/mL (IQR: 3.95–30.10), with 123 participants (=66.8%) reporting serum levels higher than normal values. Appendix A shows the distribution of serum IL-6 levels in the patients included. As reported in Table 1, patients with higher serum IL-6 levels did not differ in terms of mean age (*p* = 0.15) or gender (*p* = 0.89) or in terms of PaO_2_/FiO_2_ ratio, hemoglobin levels, and renal function. As expected, patients with higher serum levels of IL-6 reported significantly higher levels of serum-C-reactive protein (*p* = 0.001). Moreover, we failed to observe any significant differences in terms of the comorbidities analyzed (all with a *p*-value > 0.05). Finally, patients with high serum levels of IL-6 reported a significantly higher presence of pneumonia, detected with a CT scan, than their counterparts with normal serum levels (*p* = 0.048) (Table 1).

The presence of long COVID was assessed after at least one year from hospital discharge and, in the median, after 17 (range: 13–22) months. Overall, the majority reported the presence of a long COVID sign or symptom (=110/181). The most represented sign/symptom was fatigue. As reported in Figure 1 and Appendix A, compared to patients with normal serum IL-6 levels at the baseline and after adjusting our analyses for 11 potential confounders during hospitalization, higher serum IL-6 levels were associated with a doubled higher risk of long COVID (OR = 2.05; 95% CI: 1.04–4.50; *p* = 0.03). In particular, among the signs and symptoms attributable to long COVID, high serum IL-6 levels were associated with a higher incidence of mobility decline (OR = 2.55; 95% CI: 1.08–9.40; *p* = 0.02) and PTSD (OR = 2.38; 95% CI: 1.06–8.61; *p* = 0.02) (Figure 1; Appendix A). Even if higher serum IL-6 levels seemed to increase the risk of several other long COVID signs/symptoms, they were not statistically significant (*p* > 0.05), as reported in Appendix A.

## 3. Discussion

Since December 2019, when a new Coronavirus called severe acute respiratory syndrome from Coronavirus 2 (SARS-CoV-2) or (2019-nCoV) of unknown origin spread to the Chinese province of Hubei, the whole world has experienced the new pandemic of COVID-19, the epidemic disease caused by SARS-CoV-2. To rapidly diagnose and control such a highly communicable disease, suspicious individuals were isolated, and diagnostic/therapeutic procedures were developed through the analysis of epidemiological and clinical data of patients. However, due to the rapid worldwide spread of the virus, COVID-19 has become a matter of serious concern in the medical community.

Currently, the spread of the pandemic is increasingly under control thanks to the use of vaccines, and COVID-19 is managed by available antiviral drugs to improve symptoms, while supportive care, which includes oxygen and mechanical ventilation, is used for patients with more severe disease.

Mass vaccination campaigns effectively implemented in Western countries have reduced the number of hospitalizations and, especially, of deaths; however, the modalities of the evolution of the infection are still not completely clear, as well as the causes underlying the great individual variability in the host response to infection and, above all, the precise mechanisms which can cause the persistence of the symptoms known for long COVID.

Studies investigating the immune profiles during acute SARS-CoV-2 infection identified various mechanisms involved in the pathogenesis of the disease, and, in particular, a maladapted immune response profile associated with severe COVID-19 and poor clinical outcome has been documented [20]. Focusing on these aspects, Ruenjaiman et al. investigated the innate immune cell profiles in recovered COVID-19 patients at 1 and 3 months after hospital discharge in a Thai cohort of patients, and found that increased monocytes and IL-6- and TNF-α-producing cells were significantly associated with long COVID-19 symptoms [21].

Moreover, in a recently published case report on a patient with long COVID and rheumatoid arthritis, it was observed that adding the IL-6 inhibitor tocilizumab to nirmatrelvir/ritonavir could ameliorate symptoms like headache, fatigue, and brain fog, thus indirectly suggesting a possible role of IL-6 in determining those symptoms [22].

To contribute to the search for IL-6 levels as outcome predictors in COVID-19, our study was based on the observation of a population of subjects admitted to the COVID Internal Medicine Unit at the University Hospital “P. Giaccone” of Palermo, Italy, during the second Italian pandemic wave, starting from the 1 September 2020. The hospitalized subjects came from the province of Palermo and other provinces of western Sicily (Trapani and Agrigento) as a consequence of the distribution of infected subjects by the regional control room managed by the emergency service.

Our study, including almost 200 patients with previous COVID-19 and followed-up for a median of one year and half, reported overall that long COVID symptomatology is still present after so long a time, and that this condition may affect more than 50% of the population included, making this condition a public health priority.

Inflammation, particularly represented by high IL-6 levels, seems to be a fundamental factor in the pathogenesis of COVID-19 [23], as confirmed by our study. Other researchers substantially confirmed these findings in clinical and laboratory models. However, in our case study, we observed in those patients with high serum levels of IL-6 not only a significantly higher presence of pneumonia, detected with a CT scan, than their counterparts with normal serum levels, thus underling its role in the acute phase of the disease, but also a doubled higher risk of developing a form of long COVID, probably related to its involvement in long COVID pathogenesis

For example, in different parts of hamster brains, SARS-CoV-2 seems to have unique effects on the hamster olfactory system, in which the nose showed signs of extensive inflammation long after the virus could be detected [24]. Similarly, it was reported that chronic inflammation detected in the olfactory system and present after acute COVID-19 infection was associated with behavioral changes in the hamsters [24]. In this regard, our study reported that patients with higher levels of IL-6 had a significant higher risk of developing PTSD than patients with normal serum IL-6 levels. Although olfactory bulb tissue from people who recovered from COVID-19 is, as is expected, difficult to obtain, the few samples studied seem to be comparable to that of the hamsters [24].

At the same time, it should be observed that the literature regarding inflammation and long COVID in human beings is more theoretical than practical, being often based on some assumption such as the role of cytokine storms in acute phases of COVID-19 [25]. Therefore, we believe that our study is one of the few supporting, from a clinical point of view, the importance of inflammation as a basis for long COVID signs and symptoms.

Another important finding of our study is that patients with higher serum IL-6 levels had a significantly higher probability of developing mobility issues during the follow-up period, overall reinforcing the importance of inflammation in developing conditions associated with these kinds of issues, such as sarcopenia. The pathological loss of muscle mass and, finally, sarcopenia seem to be extremely important from a prognostic point of view in determining poor prognosis in people affected by COVID-19 [26], and probably in those affected by long COVID.

Altogether, our study indicates that inflammation could be a target for treating long COVID. In this sense, previous literature indicates that the management of patients with long COVID should include the promotion of healthy lifestyle habits (e.g., nutrition habits and exercise) [27] to reduce the impact of proinflammatory status, even if future intervention studies are needed for confirming these findings.

The findings of our study must be interpreted within its limitations. First, the evaluation of serum IL-6 was made only at hospital admission and during the hospital stay, and was not repeated during the follow-up period. Second, even if we clearly asked for signs and symptoms associated with COVID-19, we cannot exclude that this symptomatology could be attributable to other conditions. Finally, long COVID signs and symptoms were evaluated only using phone calls and the response rate was moderate, potentially introducing a selection bias.

## 4. Materials and Methods

### 4.1. Study Population

Patients aged >18 years and hospitalized in internal medicine or geriatrics wards from 1 September 2020 in the University Hospital (Policlinico) ‘P. Giaccone’ in Palermo, Sicily, Italy, with a diagnosis of SARS-CoV-2 infection confirmed by the investigation of SARS-CoV-2 nucleic acid on nasopharyngeal swab by means of RT-PCR were enrolled in this study [28]. No other inclusion criteria were proposed to better represent the real-life hospitalized population. The study was approved by the Local Ethical Committee during the session on 28 April 2021 (number 04/2021).

Among the 430 patients initially hospitalized, we excluded 59 deceased during hospitalization or during the follow-up period; 156 did not answer to our phone calls, 11 rejected the phone questionnaire, and 20 did not have any evaluation of serum IL-6 during the first 4 days. Therefore, 184 patients were finally included in the analysis (Figure 2).

### 4.2. Exposure: Serum IL-6 Levels

Serum samples were separated by centrifugation at 3000 rpm for 10 min. IL-6 levels were measured by electrochemiluminescence immunoassay on a Roche Cobas 8000 automated analyzer (Roche Diagnostics, Basel, Switzerland), according to the manufacturer’s procedures. The normal upper limit was 7 pg/mL, with a detection limit of 1.5 pg/mL. During hospitalization, IL-6 was measured four times: at baseline (BL); on the 4th day after hospital admission; from the 5th to the 10th day; and from the 11th to the 15th day. For the aims of this study, we considered the serum IL-6 levels measured during the first four days (exposure) and the maximum value reached (covariate).

### 4.3. Outcomes: Long COVID

The incidence of long COVID-19 symptomatology was considered as the primary outcome. We investigated, as signs or symptoms of long COVID, those based on recent systematic reviews [1,2,3,29]; i.e., neurological, respiratory, mobility impairment, heart, digestive, skin, or general signs and symptoms. All of the questions were posed as yes/no questions by phone. Psychiatric conditions were investigated using the Posttraumatic Stress Disorder Checklist-5 (PCL) for PTSD (post-traumatic stress disorder) [30] and the Hospital Anxiety and Depression Scale (HADS) for detecting anxiety and depression [31]. All of the questions were posed using phone calls.

### 4.4. Covariates and Parameters

For the aims of this study, we reported the information that, based on the literature, could affect the potential association between serum IL-6 levels and long COVID; i.e., age, gender, smoking status (actual vs. previous vs. never), and the presence of pneumonia detected using a CT scan. Among laboratory measures, we reported renal function, measured with creatinine, hemoglobin, other serum parameters of inflammation (white blood cells, C reactive protein [CRP]), and parameters of arterial blood gas exchange expressed as PaO_2_/FiO_2_ ratio (with a value less than 150 indicative for acute respiratory failure) [32]. The presence and the severity of comorbidities were investigated using the Cumulative Illness Rating Scale (CIRS) [33] that estimates the severity of pathology in each of the 13 systems, with a grade from 0 to 4.

### 4.5. Statistical Analyses

All patient records and information were anonymized and de-identified prior to the analysis. Data on continuous variables were normally distributed according to the Kolmogorov-Smirnov test and then reported as means and standard deviation values (SD) for quantitative measures and percentages for the categorical variables, by serum IL-6 levels. In cases of non-normal distribution, the data were reported as the median with IQR (interquartile) range. Levene’s test was used to test the homoscedasticity of variances and, if its assumption was violated, Welch’s ANOVA was used. *p* values were calculated using the Student’s t-test for continuous variables and the Mantel-Haenszel Chi-square test for categorical ones.

The association between serum IL-6 levels at the baseline divided in less or more than 7 pg/mL and long COVID were reported using an adjusted logistic regression and reported as odds ratios (ORs) with their 95% CI. Potential confounders were introduced in the models if they did differ between low and high serum IL-6 levels (*p*-value < 0.05) or if they were associated with the outcomes of interest using a threshold of the *p*-value of 0.10. Collinearity among factors was analyzed using the variance inflation factor (VIF) of two as a reason for exclusion, but not one of the factors was excluded for this reason.

All analyses were performed using SPSS 26.0 for Windows (SPSS Inc., Chicago, IL, USA). All statistical tests were two-tailed and statistical significance was assumed for a *p*-value of <0.05.

## 5. Conclusions

The analysis of our case series confirmed the prominent role of IL-6 levels in response to SARS-CoV-2 infection, as predictors not only of COVID-19 disease severity and unfavorable outcomes, but also long COVID development trends. If confirmed, these results would be helpful to identify subjects with SARS-CoV-2 infection who are most at risk of a persistent long COVID condition, and who should be monitored and followed-up more closely after hospital discharge.

## Figures and Tables

**Figure 1 ijms-24-01731-f001:**
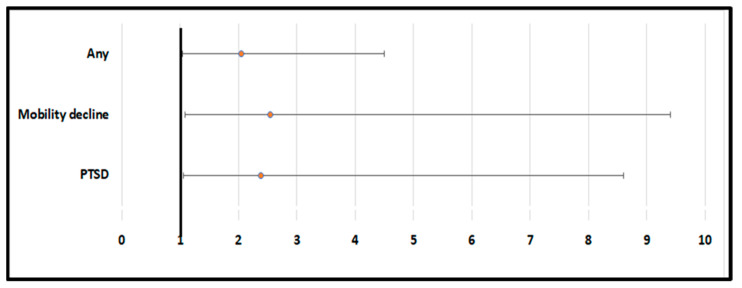
Association between baseline serum IL-6 levels and long COVID symptomatology (statistically significant results). All the data are reported as odds ratio with 95% confidence interval, calculated for elevated serum IL-6 vs. normal values and adjusted for age, sex, comorbidities (yes vs. no), smoking status (actual, previous, never), PaO_2_/FiO_2_ ratio, hemoglobin levels, renal function (all measured within the first four days of hospitalization), presence of pneumonia during hospitalization, and other serum parameters of inflammation (white cells, C reactive protein, changes during hospitalization of IL-6).

**Figure 2 ijms-24-01731-f002:**
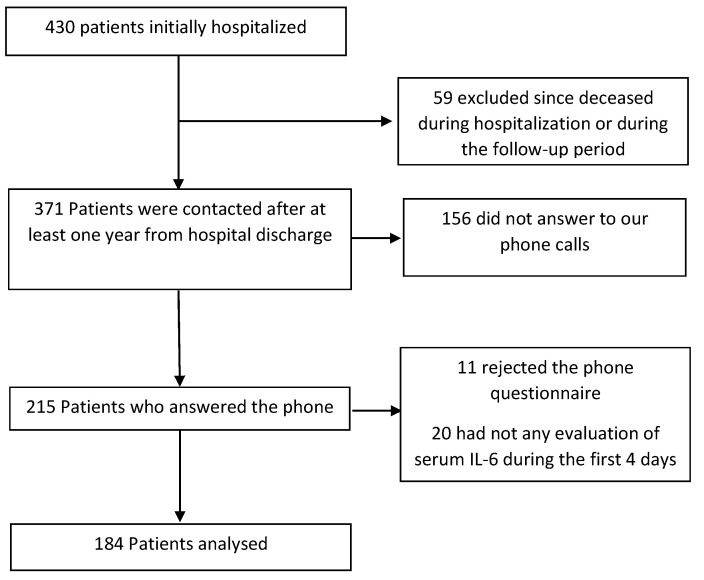
Flow chart: study population break-down by inclusion criteria.

**Table 1 ijms-24-01731-t001:** Baseline characteristics by serum IL-6 levels.

Parameter	IL-6 Normal (*n* = 58)	IL-6 Elevated (*n* = 123)	*p*-Value
Demographics			
Age (mean, SD)	60.0 (15.2)	63.1 (12.6)	0.18
Female sex (%)	48.3	47.2	0.89
Actual/previous smokers (%)	12.1	17.9	0.11
Laboratory measures			
PaO_2_/FiO_2_ ratio	314 (79)	324 (84)	0.50
White blood cells	8318 (3194)	7987 (5027)	0.65
Serum CRP levels (mg/dL)	9.82 (2.65–25.04)	56.0 (7.03–67.77)	<0.0001
Hemoglobin (g/dL)	13.9 (1.6)	13.4 (1.8)	0.12
Creatinine clearance (mL/min)	84 (25)	81 (27)	0.48
Medical conditions			
Cardiac diseases (%)	15.2	12.1	0.31
Vascular diseases (%)	57.6	53.8	0.80
Hematological diseases (%)	3.0	19.8	0.11
Respiratory diseases (%)	15.2	11.0	0.15
Eyes, ears, nose, throat, and larynx conditions (%)	6.1	7.7	0.94
Upper gastrointestinal conditions (%)	10.1	11.0	0.33
Lower gastrointestinal conditions (%)	3.0	10.9	0.60
Liver, pancreas, and biliary conditions (%)	0.0	9.9	0.17
Kidney conditions (%)	3.0	4.4	0.85
Genitourinary conditions (%)	6.1	14.3	0.43
Musculoskeletal and skin conditions (%)	11.1	4.4	0.17
Neurological conditions (%)	12.1	7.7	0.15
Endocrine (including sepsis and breast) conditions (%)	27.3	39.6	0.26
Psychiatric illnesses (including dementia) (%)	3.0	8.7	0.68
Pneumonia (%)	89.7	96.7	0.048

Data are reported as means with standard deviation (SD) for continuous variables with a normal distribution and as medians and interquartile range for those with a skewed distribution. Categorical variables were reported as percentages.

## Data Availability

Data are available upon reasonable request to the corresponding author.

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
