# Peer review of "Can Baseline IL-6 Levels Predict Long COVID in Subjects Hospitalized for SARS-CoV-2 Disease?"

_ijms, 2023, doi:10.3390/ijms24021731_

Round 1
Reviewer 1 Report
The authors reported an association between IL-6 levels at presentation and long COVID symptoms in a series of 184 COVID-19 patients. The authors noted an association between high IL-6 levels and long COVID (OR 2.05), decreased mobility (OR 2.55), and PTSD (OR 2.38). Although the causal relationship is unclear, this study has some significance in increasing our knowledge of long COVID and is worthy of publication.
Author Response
The authors reported an association between IL-6 levels at presentation and long COVID symptoms in a series of 184 COVID-19 patients. The authors noted an association between high IL-6 levels and long COVID (OR 2.05), decreased mobility (OR 2.55), and PTSD (OR 2.38). Although the causal relationship is unclear, this study has some significance in increasing our knowledge of long COVID and is worthy of publication.
R: We would like to sincerely thank the Reviewer 1 for the appreciation of our work that we tried to further improve using the Reviewers’ comments.
Reviewer 2 Report
In this manuscript, the authors analyzed the levels of IL-6 in the sera of COVID-19 patients at day 4 after exposure and found that the elevation of IL-6 may be used as a marker to predict the incidence of long COVID, especially mobility decline and PTSD. These results are encouraging since IL-6R could be a target for therapy to prevent the occurrence of long-COVID symptoms.
Please clarify the time point of the level of IL-6 in this study. The author mentioned in the abstract that the level of IL-6 was at day 4 after exposure. Was day 4 the time point of the 4th day after admission into hospital? Or the day 4 patient having COVID related symptoms? Or the time which the patient infected by COVID-19 virus (exposure to virus)?
Based on the last exclusion criteria in Figure 1, 11 patients rejected the phone questionnaire and no IL-6 data of 23 patients, please check the total patients in this studies is 181 (215-11-23=181) or 184?
As a reference for readers and clinicians, it is better to include all the IL-6 individual values in the manuscript, a column figure of scatter plot to show the individual values of IL-6 from all included patients with high and normal IL-6.
There are published reports which indicated that IL-6 level may be used to predict the occurrence of severe COVID-19 (J Med Virol. 2020; 92(7): 791–796), respiratory failure (Clin Infect Dis. 2020 ;71(8):1937-1942) or long-term neuropsychiatric symptoms of VOVID-19 (Psychoneuroendocrinology. 2021; 131: 105295). Please modify the description of the 1st sentence in the section of Introduction and include the related information from these references in the manuscript.
Author Response
Please clarify the time point of the level of IL-6 in this study. The author mentioned in the abstract that the level of IL-6 was at day 4 after exposure. Was day 4 the time point of the 4th day after admission into hospital? Or the day 4 patient having COVID related symptoms? Or the time which the patient infected by COVID-19 virus (exposure to virus)?
R: Thank you for the comment. We have clarified that the measurement of the serum IL6 levels was made four days after the hospital admission.
Based on the last exclusion criteria in Figure 1, 11 patients rejected the phone questionnaire and no IL-6 data of 23 patients, please check the total patients in this studies is 181 (215-11-23=181) or 184?
R: Sorry for the typo. Overall, 20 and not 23 had not serum IL6 measurement during the first 4 days. We have corrected the Figure and the manuscript accordingly.
As a reference for readers and clinicians, it is better to include all the IL-6 individual values in the manuscript, a column figure of scatter plot to show the individual values of IL-6 from all included patients with high and normal IL-6.
R: Good point. We added this Figure in the Supplementary Material, as suggested.
There are published reports which indicated that IL-6 level may be used to predict the occurrence of severe COVID-19 (J Med Virol. 2020; 92(7): 791–796), respiratory failure (Clin Infect Dis. 2020 ;71(8):1937-1942) or long-term neuropsychiatric symptoms of VOVID-19 (Psychoneuroendocrinology. 2021; 131: 105295). Please modify the description of the 1st sentence in the section of Introduction and include the related information from these references in the manuscript.
R: Thank you for this comment. We added this paragraph in the Introduction section, as follows:
“Some studies indicated that IL-6 levels may be used to predict the occurrence of severe COVID-19 [8], respiratory failure[9] or long-term neuropsychiatric symptoms of COVID-19 [10].”
Reviewer 3 Report
The manuscript is written well. It is very easy to follow. However, authors should add more information about IL-6 function during infection in the introduction section. Also, they should emphasize the significance of the study in discussion and conclusion sections. Moreover, the first paragraph of results section and Figure 1 should be moved into the first section of Materials and Methods section.
Author Response
The manuscript is written well. It is very easy to follow. However, authors should add more information about IL-6 function during infection in the introduction section. Also, they should emphasize the significance of the study in discussion and conclusion sections.
- We thank the reviewer for the suggestions; in the introduction section we added more information about IL-6 function during infection with the following sentences and the corresponding references: “During the evolution of COVID-19 clinical picture the secretion of multiple cytokines is closely related to the development of clinical symptoms and in particular IL-6 can be involved into vascular leakage, activation of complement and the coagulation cascade, leading to the characteristic symptoms of severe forms, such as those that can develop a diffuse intravascular coagulation (DIC) [11, 12]. IL-6 is also likely to cause cardiomyopathy by promoting myocardial dysfunction, which is often observed in patients with critical forms of COVID-19 [13]. In addition, activation of endothelial cells and endothelial dysfunction can lead to capillary leakage, hypotension, and coagulopathy [14]”.
We then emphasized the significance of the study in the discussion section with the sentence:
“But in our case study we observed in those patients with high serum levels of IL-6 not only a significantly higher presence of pneumonia, detected with CT scan, than their counterparts with normal serum levels, thus underling its role in the acute phase of the disease, but also a doubled higher risk of developing a form of long COVID, probably related to its involvement in long COVID pathogenesis”.
In the conclusions the sentence
“Further studies are necessary to confirm these results to identify subjects with SARS-CoV-2 infection most at risk of persistence of a long COVID condition” has been changed into: “If confirmed these results would be helpful to identify subjects with SARS-CoV-2 infection most at risk of persistence of a long COVID condition who should be monitored and followed up more closely after hospital discharge”.
Moreover, the first paragraph of results section and Figure 1 should be moved into the first section of Materials and Methods section.
R: Done.